# Molecular Mechanisms Underlying the Effects of Olive Oil Triterpenic Acids in Obesity and Related Diseases

**DOI:** 10.3390/nu14081606

**Published:** 2022-04-12

**Authors:** Carmen M. Claro-Cala, Francesc Jiménez-Altayó, Sebastián Zagmutt, Rosalia Rodriguez-Rodriguez

**Affiliations:** 1Departament of Pharmacology, Pediatríc y Radiology, Faculty of Medicine, University of Seville, 41009 Seville, Spain; cmclaro@us.es; 2Departament de Farmacologia, de Terapèutica i de Toxicologia, Facultat de Medicina, Institut de Neurociències, Universitat Autònoma de Barcelona, 08193 Bellaterra, Spain; francesc.jimenez@uab.cat; 3Basic Sciences Department, Faculty of Medicine and Health Sciences, Universitat Internacional de Catalunya, 08195 Sant Cugat del Valles, Spain; szagmutt@uic.es

**Keywords:** triterpenes, olive oil, obesity, insulin resistance, adiposity, vascular diseases, oleanolic acid

## Abstract

Dietary components exert protective effects against obesity and related metabolic and cardiovascular disturbances by interfering with the molecular pathways leading to these pathologies. Dietary biomolecules are currently promising strategies to help in the management of obesity and metabolic syndrome, which are still unmet medical issues. Olive oil, a key component of the Mediterranean diet, provides an exceptional lipid matrix highly rich in bioactive molecules. Among them, the pentacyclic triterpenic acids (i.e., oleanolic acid) have gained clinical relevance in the last decade due to their wide range of biological actions, particularly in terms of vascular function, obesity and insulin resistance. Considering the promising effects of these triterpenic compounds as nutraceuticals and components of functional foods against obesity and associated complications, the aim of our review is to decipher and discuss the main molecular mechanisms underlying these effects driven by olive oil triterpenes, in particular by oleanolic acid. Special attention is paid to their signaling and targets related to glucose and insulin homeostasis, lipid metabolism, adiposity and cardiovascular dysfunction in obesity. Our study is aimed at providing a better understanding of the impact of dietary components of olive oil in the long-term management of obesity and metabolic syndrome in humans.

## 1. Introduction

Obesity arises from an imbalance between energy intake and expenditure, leading to an excessive accumulation of fat and inflammation that correlates with a dysfunctional regulation in metabolic organs or tissues and the so-called metabolic syndrome [1]. This disease represents a global epidemic and a major risk factor for type-2 diabetes, cardiovascular diseases, cancer and, more recently, COVID-19 outcomes [2,3]. If the global trend continues, worldwide obesity and overweight prevalence in 2025 will be around 21% in women and 18% in men, causing a serious burden on public health, with severe economic and social consequences [4]. The alarming increase in obesity and related complications indicates that it is still an unmet medical issue that requires new therapeutic strategies. 

Currently, several drugs have been developed to target individual components of obesity and metabolic syndrome, but unfortunately, none of them have yet been demonstrated to solve the root causes or ameliorate the long-term disease. Given the limited success of pharmacological interventions against obesity, physical activity and dietary-based approaches are an integral part of obesity management [5,6,7]. Particularly, certain components of the diet exert protective effects against obesity and related metabolic and cardiovascular disturbances by interfering with the molecular pathways implicated in these pathologies. These dietary biomolecules are promising therapeutic strategies that may be employed to supplement other dietary or pharmacological products as nutraceuticals to manage obesity and metabolic syndrome [8]. 

Olive oil, a key component of the Mediterranean diet, is partly responsible for the health-promoting effects of this dietary pattern [9,10,11], providing an exceptional lipid matrix rich in molecules with different bioactive chemical entities. The pentacyclic triterpenic acids (oleanolic and maslinic) and dialcohols (uvaol and erythrodiol) have gained clinical relevance in the last decade due to their wide range of biological actions [12,13]. The triterpenic acids and diols are differentiated based on the functional group present at the C-17 position. Maslinic acid has two vicinal hydroxyl groups at the C-2 and C-3 positions besides the carboxyl group (Figure 1), whereas erythrodiol and uvaol present two hydroxyl groups in remote positions and differ depending on the methyl group location (Figure 1). The concentration range of these triterpenoids reaches values up to 400 mg/kg in the skin of olive fruit, whereas concentrations in olive oil are 8.15–85 mg/kg for dialcohol content and between 8.9–112.36 mg/kg for triterpenic acids [14,15]. Both olive oil and pomace olive oil (the residue that remains after mechanical extraction of virgin olive oil) are important sources of these pentacyclic triterpenoids [14].

The beneficial effects of olive oil triterpenes in health have been mostly attributed to oleanolic acid and, to a lesser extent, maslinic acid, in terms of vascular dysfunction, obesity, insulin resistance, and cancer [12,13,16,17]. The wide distribution of oleanolic acid and related triterpenic acids in medicinal plants has also contributed to the elucidation of their biological actions and molecular mechanisms involved in these functions [12]. As a result, functional olive oil enriched in oleanolic acid is currently being investigated in a clinical trial to prevent type 2 diabetes [18], but most of the investigations are still at a pre-clinical stage. Considering the promising effects of these triterpenic compounds as nutraceuticals against obesity and associated disturbances, such as diabetes and cardiovascular diseases, the aim of our review is to decipher and discuss the main molecular mechanisms underlying these effects driven by olive oil triterpenes, in particular oleanolic acid. Special attention will be paid to their signaling pathways and targets related to body weight, feeding, glucose and insulin homeostasis as well as to adiposity, inflammation, oxidative stress and vascular dysfunction in obesity. Therefore, this study is aimed at providing a better understanding of the impact of dietary components of olive oil in the long-term management of obesity and metabolic syndrome in humans.

## 2. Molecular Mechanisms Underlying the Anti-Obesity Effects of Triterpenic Acids

Olive oil triterpenic acids, particularly oleanolic acid, have shown beneficial effects in obesity and related complications in vitro and in vivo animal models. Although these properties are clear in terms of body weight, glycemia, insulin sensitivity, adiposity and cardiovascular function, the mechanisms underlying these actions are complex and multitargeted. In the following subsections, we discuss the main investigations exploring these molecular mechanisms, which are directly related to glucose and insulin homeostasis, lipid homeostasis and cardiovascular function, despite the lack of studies on the specific pathways involved in feeding behavior. 

### 2.1. Effects of Triterpenic Acids on Glucose Homeostasis and Insulin Resistance

Insulin is a key regulator of glucose homeostasis through main regulatory actions in peripheral organs and the brain. In particular, in response to diet, insulin secretion from beta cells drives, among others, liver glucose and glycogen production, glucose uptake in skeletal muscle, lipolysis attenuation in adipose tissue, central reduction of feeding, vasodilatation and excretion of glucose (Figure 2) [19]. In these organs or tissues, insulin actions are mostly regulated via activation of the insulin receptor (IR) and the tyrosine (Tyr) phosphorylation of insulin receptor substrate (IRS) proteins and other signaling molecules (e.g., Shc) (Figure 2). The phosphorylated IRS act as docking proteins for other proteins, such as phosphoinositide 3-kinase (PI3K), leading to the activation of downstream effectors (protein kinase B (Akt) and atypical protein kinase C (aPKC)) that regulate glucose transport (e.g., glucose transporter type 4 (GLUT4) translocation), glycogen and lipid synthesis, and other signals related to proliferation. Other cellular insults, such as mitochondrial dysfunction, reactive oxygen species (ROS) production and inflammatory signals can disrupt these insulin-dependent molecular pathways [20]. In obesity and type 2 diabetes, on the one hand, increased adiposity leads to a decreased production of the adipokine adiponectin with systemic inflammation and higher lipolysis (Figure 2). This process contributes to the ectopic accumulation of lipids in the liver and skeletal muscle with subsequent attenuation of glucose uptake in muscle and higher gluconeogenesis and lipogenesis in the liver with lower glycogen synthesis (Figure 2) [19]. In addition to lower secretion of pancreatic insulin, insulin resistance is also evident in the brain (attenuating the anorexigenic action of this hormone), kidney and vascular endothelium (Figure 2).

Among olive oil triterpenes, oleanolic acid is known to regulate insulin action and glucose metabolism. Administration of oleanolic acid decreased plasmatic levels of glucose in both hyperglycemic rats [21] and diet-induced obese mice with a significant attenuation of insulin resistance, body weight and adiposity [22,23,24]. Oleanolic acid is also able to ameliorate blood glucose and insulin in diabetic mice [25]. The mechanisms by which oleanolic acid acts on insulin and glucose homeostasis seem to be complex and multitargeted. A principal mechanism involves liver glucose production and uptake (Figure 2). Intraperitoneal administration of oleanolic acid for 2 weeks led to a restoration of liver gluconeogenesis, mitochondrial dysfunction and inflammation in db/db diabetic mice [25] (Figure 2). These effects were coupled with a reduction in fasting glycemia, fasting insulinemia and body weight. Alterations in glycogen levels produced by liver and muscle were also shown by oral administration of oleanolic acid in high-fat diet-induced prediabetic rats [26]. In addition to the action of oleanolic acid on glucose production, this triterpenic acid is able to induce a dual agonist action on peroxisome proliferator-activated receptor gamma/alpha ((PPAR)γ/α) in adipocytes and GLUT4 translocation in myoblasts (Figure 2), indicating a mechanism based on adipogenesis and glucose uptake in peripheral tissues to improve type 2 diabetes [27].

Besides mechanisms based on glucose production and uptake, oleanolic acid also affects the pancreatic secretion of insulin. A study by Teodoro et al. [28] examined the insulin secretagogue properties of oleanolic acid using rat glucose-responsive pancreatic cell lines and isolated islets. In these in vitro models, oleanolic acid enhanced insulin secretion and content at basal and after exposure to different glucose concentrations. In the cell lines, the effects of oleanolic acid on insulin secretion were comparable to those led by the sulphonylurea tolbutamide and also by the incretin mimetic Exendin-4. However, oleanolic acid was unable to emulate the same mechanism of action of these drugs, namely stimulation of intracellular calcium release and cyclic adenosine monophosphate (cAMP) production by beta-cells, suggesting another mechanism involved in insulin secretion in response to the triterpenic acid. The major drawback of this study is that the concentration of oleanolic acid used to induce effects on insulin secretion in vitro was much higher than those reported in rodent plasma after oral administration of oleanolic acid to lower plasma glycemia in vivo [28]. The direct effects of oleanolic acid on the pancreatic secretion of insulin need further in vivo evaluation. 

Oleanolic acid is also affecting specific inflammatory targets to improve insulin sensitivity in both liver and adipose tissue (Figure 2). Particularly, pro-inflammatory cytokines and some lipogenic genes in liver and adipose tissue were attenuated by oleanolic acid via intraperitoneal [25] and oral administration in either isolated form [22] or in a pomace olive oil concentrated in oleanolic and maslinic acid (POCTA) [23] in diet-induced obese mice. Anti-inflammatory effects may improve insulin sensitivity in these tissues leading to a better oral glucose test, insulin tolerance test, adiposity attenuation and a substantial reduction in body weight gain, despite no changes in food intake [23]. Additionally, the anti-inflammatory effects of oleanolic acid contribute to alleviating rat carotid artery injury induced by hyperglycemia in diabetic rats by inhibition of the nucleotide-binding domain and leucine-rich repeat protein-3 (NLRP3) inflammasome signaling pathways in the neointima [29]. 

The effects of oleanolic acid on insulin resistance are also linked to antioxidant mechanisms in the liver (Figure 2). Two studies performed by Nyakudya et al. [30,31] showed the protective effects of neonatal administration of oleanolic acid for 16 weeks in high fructose fed rats, in terms of glycemia and glucose tolerance test improvement compared to control animals, results that were associated with the antioxidant action of the triterpenic acid in the liver. In line with these results, Su et al. [32] found that oral administration of oleanolic acid reduces polychlorinated biphenyls (PCBs)-induced adiposity and insulin resistance via hepatocyte nuclear factor-1-beta (HNF1b)-mediated regulation of redox and PPARγ signaling in adipose tissue in mice. The antioxidant effect of oleanolic acid in the liver and adipose tissue was also demonstrated by Wang et al. [33] using nanoencapsulated forms of the triterpene by oral gavage, showing attenuation of malondialdehyde (MDA) and nitric oxide (NO) with the restored activity of superoxide dismutase (SOD) and catalase (CAT) in blood samples (Figure 2). Interestingly, nano-oleanolic acid was able to reduce high fat and fructose diet-induced lipid accumulation in the liver and improve mitochondrial and endoplasmic reticulum (ER) structural integrity in the liver and pancreas. The antioxidant-dependent mechanisms in the anti-diabetic effects of oleanolic acid have been also evidenced in type 2 diabetic rats, finding an attenuation in oxidative stress and ER stress in the kidney of these animals in response to the oral administration of oleanolic acid in combination with N-acetyltransferase [34]. As a result of these effects, this therapy increased nephrin and endothelial selective adhesion molecules, leading to an attenuation of apoptosis and diabetic nephropathy. 

The beneficial effect of oleanolic acid on animal models of diabetes has also been related to changes in the IRS-1/PI3K/Akt signaling in metabolic organs (Figure 2). In particular, oral administration of oleanolic acid for 10 weeks restored this pathway in the adipose tissue of adipo-IR rats (a diabetes model induced by fructose) [35]. Treatment with oleanolic acid upregulated mRNA expression of the insulin receptors, IRS-1, PI3K and Akt in white adipose tissue (Figure 2). Moreover, the triterpene restored the expression ratio of phospho-IRS-1 (pIRS-1/IRS1) and phospho-Akt (pAkt)/Akt, which was significantly altered in fructose-induced diabetic rats, to the expression levels of the control group [35]. The involvement of this pathway in the anti-diabetic effect of oleanolic acid was also reported by Wang et al. [36] when exploring the synergic action of oleanolic acid with metformin to treat diabetes in db/db diabetic mice. The anti-hyperglycemic action of metformin is based on a decrease of liver glucose production by adenosine monophosphate activated protein kinase (AMPK)-dependent and independent mechanisms [37]. The combination therapy of both metformin and oleanolic induced a significant reduction of blood glucose levels and improved liver function in diabetic mice compared to the monotherapy, involving AMPK-dependent pathways [36]. In particular, oleanolic acid plus metformin increased phosphorylation of Akt, PI3K, AMPK and acetyl-coenzyme A carboxylase (ACC), and increased liver glycogen, whereas it decreased the expression of proteins involved in liver gluconeogenesis and glycolysis (i.e., glucose-6-phosphatase (G6Pase) and phosphoenolpyruvate carboxykinase (PEPCK)) compared to those levels under monotherapy, and attenuated the liver phosphorylation of mammalian target of rapamycin (mTOR) and cyclic adenosine monophosphate (cAMP)-response element binding protein (CREB) [36]. The complexity underlying the pathogenesis of diabetes, particularly in obesity suggests that combination therapy, including not only well-known anti-diabetic drugs, such as metformin but also new approaches, such as oleanolic acid, may be more effective in treating these metabolic disturbances. Interestingly, co-administration of metformin with oleanolic acid results in a net inhibition of gluconeogenesis and glycogenolysis and a reduction of liver glucose production at more efficient levels compared to these drugs in monotherapy, indicating the potential benefit of this strategy against metabolic disorders. 

The effect of oleanolic acid on glucose and insulin homeostasis has also been attributed to its regulatory action in the activity of tyrosine phosphatases downstream IRS-1 [38,39]. On the one hand, Bu et al. [38] used a model for high-throughput screening of protein tyrosine phosphatase 2 (Shp2) enhancers, identifying oleanolic acid as a substrate to enhance Shp2 activity. Shp2 is a tyrosine phosphatase expressed in several tissues, and its deletion in neurons, muscle, or pancreatic beta-cells led to an obesogenic and diabetic phenotype in mice. The protective effects of oleanolic acid on glucose and insulin homeostasis could be related to a restoration of Shp2 activity. The inhibitory action of oleanolic acid and related compounds against other tyrosine phosphatases, such as protein tyrosine phosphatase 1B (PTP-1B) was also shown in vitro and in silico, as a potential mechanism underlying the glycemic-lowering action of intragastric oleanolic acid administration in streptozotocin-induced diabetic rats [39]. Nevertheless, these observations should be confirmed by additional in vivo experiments.

To sum up, the molecular mechanisms underlying the effects of oleanolic acid in glucose and insulin homeostasis are complex and multitargeted. These mechanisms mainly involve pathways related to liver gluconeogenesis and glycogenolysis, IRS-1/PI3K/Akt in adipose tissue and liver, and glucose uptake in the muscle. The beneficial action of olive oil triterpenic acids on glucose and insulin signaling are consistent and concur with body weight loss, but with a lack of information about the feeding pattern and hypothalamic action of insulin. New approaches combining anti-diabetic drugs with olive oil triterpenic acids are remarkable and promising strategies to fight against these metabolic pathologies. However, there are still many open questions regarding these specific mechanisms, and more in vivo experiments with models of obesity and metabolic syndrome are needed regarding insulin homeostasis and food intake. 

### 2.2. Effects of Triterpenic Acids on Lipid Homeostasis, Adiposity and Inflammation

In metabolic syndrome, the liver shows increased expression of transcription factors involved in adipogenesis, fatty acid synthesis, and insulin resistance. Uncontrolled upregulation of sterol regulatory element binding protein 1 (SREBP-1) leads to increased liver lipogenesis that induces insulin resistance [40]. In a fructose-induced study of rat fatty liver, oleanolic acid suppressed the expression of SREBP-1c and SREBP-1c-responsive genes and the synthesis of lipogenic enzymes, such as fatty acid synthetase (FAS), ACC and liver stearoyl-coenzyme-desaturase-1 (SCD1) [41,42] (Figure 3). Furthermore, as a lipogenic mediator, PPARα is involved in fatty acid synthesis, degradation, storage, transport, ketogenesis, and lipoprotein metabolism [43]. In studies with oleanolic acid [42], authors reported an improvement in liver lipid metabolism through PPARα-mediated pathways [35,44], showing an anti-inflammatory and anti-obesity effect in obese mice fed a high-fat diet [22] (Figure 3).

On the other hand, adipose tissue development is regulated by several transcription factors and cell cycle regulators. Among them, the transcription factors that are mainly involved in abnormal adipogenesis are cytosine-cytosine-adenosine-adenosine-thymidine (CCAAT) enhancer-binding protein alpha (C/EBPα) and PPARγ (both expressed in white and brown adipocytes), which regulate the differentiation of adipocytes. An in vitro study in 3T3-L1 preadipocytes [45] reported that ursolic and oleanolic acid are potent anti-obesity agents, as they interfere in different stages of adipogenesis, lipolysis, and fatty acid oxidation in adipose tissues by down-regulating cellular expression of C/EBPα and PPARγ, and by promoting anti-adipogenic pathways and kinases (mitogen-activated protein kinase (MAPK), AMPK, and cAMP-dependent protein kinase A (PKA)) (Figure 3). Furthermore, the results obtained in the study by Pérez-Jiménez et al. [46] demonstrated that maslinic acid also produced a decrease in cytoplasmic triglyceride droplets, triglycerides, and transcription factors related to the adipogenesis process (PPARγ, aP2) in 3T3-L1 cells. In particular, this study shows that the increase in intracellular Ca^2+^ concentration produced by maslinic acid could also underlie the anti-adipogenic effect of this triterpene (Figure 3).

Furthermore, adipose tissue secretes bioactive peptides, such as adipokines, which act locally (autocrine/paracrine) and systemically (endocrine). The endocrine effect of adipose tissue is disrupted in obesity leading to dyslipidemia, prothrombotic and pro-inflammatory effects, hyperglycemia, and insulin resistance [47]. In particular, oleanolic acid was found to reduce visfatin, a fat-specific pro-inflammatory adipokine, in adipocytes through the inactivation of PPARγ [45]. Furthermore, oleanolic acid inhibited perilipin A expression in adipocytes, an effect that has been associated with protection against diet-induced obesity [12,48].

Obesity-related metabolic diseases show increased fat accumulation, in which macrophages play an important role in the progression of low-grade chronic inflammation, led by hypertrophied adipose tissue and immune cells [49]. The highly pro-inflammatory macrophage M1 phenotype, promoted by B-cell and T-cell activation, constitutes the crown structure around dying adipocytes in obese individuals [50], while lean white adipose tissue is dominated by the M2-type macrophage population, an anti-inflammatory environment associated with the Th2-type immune response. The anti-inflammatory environment of M2-type macrophages is due to their secretion of anti-inflammatory cytokines, such as arginine-1, mannose receptor C-type 1 (MRC1), and interleukin 10 (IL-10) [50]. Li et al. [51], in agreement with previous studies [22], demonstrated that oleanolic acid attenuates the inflammation of adipose tissue by diverting macrophage infiltration and polarization to the anti-inflammatory M2 phenotype, reducing the M1/M2 ratio and decreasing their size in a mouse model of obesity. Moreover, the results showed that M1 macrophage-derived pro-inflammatory adipocytokine markers, such as inducible nitric oxide synthase (iNOS), interleukin 6 (IL-6) and tumor necrosis factor alpha (TNFα) [52,53] from adipose tissues and chemokines (i.e., monocyte chemoattractant protein-1 (MCP1)) were significantly reduced in epididymal white fat of mice compared to vehicle-treated mice [51] (Figure 3). These results were verified by performing a similar assay in interferon gamma (IFN-γ)/lipopolysaccharide (LPS)-stimulated RAW264 cells. 

On the other hand, the anti-inflammatory effect of oleanolic acid on the pro-inflammatory activation of M1 macrophages due to oxidative stress is mediated by activation of the NLRP3 inflammasome in RAW264.7 cells, bone-marrow-derived macrophages and mice white adipose tissue [17,51]. In addition, oleanolic acid led to an attenuation of caspase-1 expression and inflammatory cytokines, such as interleukin 1beta (IL-1β), interleukin 18 (IL-18), and ROS production. Certain ROS act as second messengers to propagate inflammatory signals and are produced by different sources, such as peroxisomal fatty acid metabolism, phagocytic cells, cytochrome p450, or the mitochondrial respiratory chain [54]. In particular, voltage-dependent anion channels abundant in mitochondrial membrane proteins (VDAC) are critical for ROS production; oleanolic acid treatment is observed to down-regulate VDACs [51].

Another main mechanism underlying the anti-inflammatory properties of oleanolic acid is via AMPK phosphorylation in the inflammatory state. Matumba et al. [55] reported a reduced expression of IL-6 and TNFα genes mediated by increased AMPK expression in the skeletal muscle of rats fed a high-fructose diet and treated with oleanolic acid. Specifically, IL-6 is synthesized by a wide number of cells, such as adipocytes and macrophages to fibroblasts and skeletal muscle cells and is strongly associated with obesity and insulin resistance [56]. In vitro studies reported that oleanolic acid could stimulate AMPK phosphorylation in THP-1 macrophages [17,57]. 

Finally, the antioxidant properties of oleanolic acid also play a role in neuroinflammation. In fact, obesity alters insulin signaling pathways and appears to be responsible for the neuroinflammatory process and cognitive impairment [58]. In this sense, oleanolic acid showed beneficial actions on neuroinflammation in LPS-activated BV2 microglial cells, reduced the release of IL-1, IL-6, TNFα and nitrogen oxide (NO), and induced a modest inhibition of ROS production. These effects were associated with a down-regulation of genes encoding cytokines and inducible NO synthase (iNOS), as well as restoration of glutathione levels [59]. These results are consistent with previous studies [60] in PC12 cells exposed to oleanolic or ursolic acid, showing antioxidant and anti-inflammatory effects through glutathione (GSH)-free treatment, increased SOD and catalase activity, and decreased expression of IL-6 and TNFα expression. Specifically, the study by Zhang et al. [61] showed cognitive improvements by reducing the production of the inflammatory cytokines TNFα and IL-1β in the hippocampal regions of adolescent rats with Alzheimer’s disease, suggesting that oleanolic acid may ameliorate the progression of this disease by alleviating neurotoxic injury caused by astrocyte activation.

### 2.3. Effects of Triterpenic Acids on Cardiovascular Alterations Associated with Obesity

Obesity leads to numerous vascular changes that can alter cardiovascular homeostasis. Solid evidence of the potential application of triterpenic acids to combat cardiovascular disease was reported in various pioneer studies in the early 2000s. Somova et al. [62] showed that treatment with oleanolic acid and ursolic acid, two triterpenoids contained in ‘orujo’ pomace olive oil, modulated lipid profile and blood glucose levels favorably, decreased heart rate, and prevented the development of severe hypertension in Dahl salt-sensitive hypertensive rats. In addition, oleanolic acid and erythrodiol relaxed the rat aorta by a mechanism related to the endothelial production of NO [63]. Consistently, these compounds, and other components of pomace olive oil, such as maslinic acid and uvaol, evoked concentration-dependent relaxations in the aorta from spontaneously hypertensive rats [64]. Of note, oleanolic acid was the most potent vasodilator, which suggests that differences in the specific functional groups contained in the olive oil triterpenoids’ chemical structure could be determinants of the magnitude of the vasodilator response [64]. NO is of fundamental importance as a modulator of vascular tone, and loss of NO bioavailability as a consequence of, for instance, reduced synthesis or increased NO scavenging by free oxygen and nitrogen radicals participates in obesity-related endothelial dysfunction [13,65]. Mounting evidence indicates that triterpenic acids can exert potential beneficial actions on obesity-induced endothelial dysfunction by increasing NO bioavailability. Thus, potentiation of the NO-induced vasodilation exerted by triterpenic acids was effective to improve endothelial dysfunction in aortas from high fat diet-fed spontaneously hypertensive rats after chronic intake of diets rich in pomace olive oil, an effect that was associated with increased endothelial NO synthase (eNOS) expression [66]. 

Among all triterpenoids contained in olive oil, oleanolic acid is the most extensively studied compound for health effects linked to its vasoactive properties. Mechanistically, as mentioned above, oleanolic acid induces relaxation by promoting the release of NO from the endothelium, a process that does not require endothelial calcium and involves PI3K-dependent phosphorylation of Akt-Serine (Ser) (473) followed by phosphorylation of eNOS-Ser (1177) in human umbilical vein endothelial cells (HUVECs) [67] (Figure 4). Oleanolic acid was protective against high glucose-reduced NO production and Akt-Ser (473) and eNOS-Ser (1177) phosphorylation in HUVECs by a mechanism involving PPAR delta (PPARδ) activation, since a specific antagonist of PPARδ, selective PPARδ antagonist (GSK0660), abolished these effects [68]. In addition, in bovine aortic endothelial cells, oleanolic acid increased PPARδ activity and induced the overexpression of PPARδ target genes, pyruvate dehydrogenase kinase 4 (PDK4), adipose differentiation-related protein and angiopoietin-like protein 4 (ANGPTL4), and these effects were inhibited by GSK0660 [68]. In the same study, oleanolic acid prevented high glucose-induced endothelial dysfunction in the Sprague-Dawley rat aorta and the effect that was inhibited by GSK0660, confirming the involvement of PPARδ on the effects of this triterpenic acid [68] (Figure 4). 

On the other hand, some studies suggest that oleanolic acid can also act through endothelium-independent routes to exert its vasoprotective actions (Figure 4). Oleanolic acid induces cyclooxygenase-2-dependent prostacyclin release in human vascular smooth muscle cells [69]. In addition, relaxations induced by an oleanolic acid synthetic derivative were mediated either by activation of endothelial cyclooxygenase or endothelium-independent smooth muscle adenosine 5’-triphosphate (ATP)-dependent and voltage-activated K^+^ channels in the aorta and small mesenteric arteries from Wistar normotensive rats [70] (Figure 4). Altogether, these results demonstrate that oleanolic acid has a multitarget mode of action in vessels, which provides molecular insights into the promising cardiovascular benefits of this compound.

The effects of triterpenic acid-enriched diets on some of the altered cardiovascular parameters associated with obesity have been evaluated in various experimental studies. A ten week dietary intervention with pomace olive oil with high concentrations of oleanolic and maslinic acid in high-fat diet-fed mice led to a substantial reduction in body weight, insulin resistance, and adipose tissue inflammation [23]. Interestingly, this dietary intervention partially restored high-fat diet-induced aortic vasodilator dysfunction by potentiating NO-independent relaxations, as well as prevented the increase in phenylephrine (α1 adrenergic receptor agonist) contractions [23]. Similar results were obtained in the aortas of obese Zucker rats, where 8 weeks of daily supplementation with tomato-based Mediterranean sofrito (12% extra virgin olive oil) attenuated phenylephrine-induced contractions [71,72]. This effect was associated with preservation of the NO influence on vasoconstriction, likely because of attenuation of oxidative distress. 

High blood pressure is associated with a greater risk of developing cardiovascular disease. In high-fat diet fed normotensive Sprague-Dawley rats, asiatic acid (20 mg/kg/day; orally (p.o.); 3 weeks) evoked pressure-lowering effects that were associated with downregulation of the renin-angiotensin pathway [73]. In small mesenteric arteries from spontaneously hypertensive rats on a high-fat diet, chronic treatment (12 weeks) with pomace olive oil-enriched diets (up to 800 parts per million of oleanolic acid; p.o.) ameliorated endothelial function by improving both NO and endothelium-derived hyperpolarizing factor (EDHF) pathways [74]. However, this improvement in vascular function was not accompanied by changes in blood pressure either in this study [74] or in a previous one [66]. In contrast, in spontaneously hypertensive rats fed a normal diet, a shorter duration (8 weeks) of pomace olive oil concentrated in triterpenic acids (56.8 mg/kg/day oleanolic acid; 38.0 mg/kg/day maslinic acid; p.o.) was able to reduce blood pressure values in conjunction with improved cardiac hemodynamics and decreased relative kidney and lung weights [75]. In addition, this treatment was associated with an improvement in endothelium-dependent relaxation, enhancement of eNOS expression, and decreased levels of TNFα, transforming growth factor beta (TGFβ), and collagen I [75]. Furthermore, chronic administration (60 mg/kg/day; intraperitoneal injection (i.p.); 6 weeks) of olive leaf extracts from different subspecies and isolated oleanolic acid prevented the development of hypertension in hypertensive Dahl salt-sensitive rats [76]. Overall, these results suggest that triterpenic acids exert antihypertensive effects, but it is reasonable to think that these actions might be more difficult to observe when obesity and hypertension coincide. Aspects, such as the specific triterpenoid/s used, dosage, route of administration, and duration of treatment are likely determinant factors. Because obesity and hypertension often coexist in the same patient, it would be likely necessary to redesign therapeutic approaches with triterpenic acids for the treatment of obesity-associated hypertension to improve treatment outcomes. 

## 3. Bioavailability and Clinical Potential in Obesity

Triterpenes have a relatively low solubility in water and oil and do not permeate efficiently across cell membranes, which limits their bioavailability and therapeutic potential. Despite its unfavorable pharmacokinetics, triterpenic acids reach the plasma compartment in humans [77,78,79]. In fact, intact forms of triterpenic acids are present in tissues after oral intake (0.5 g triterpenes/99.5 g diet; 4 or 8 weeks) in mice, including in the brain [80]. Nonetheless, the liver is the major organ for triterpene storage and/or metabolism [80], which suggests that: (i) the bioavailability of these compounds could be improved; and (ii) despite a relatively safe profile, hepatotoxicity should be monitored, particularly in chronic treatments.

The bioavailability of these compounds depends on numerous factors, but gastrointestinal absorption and metabolism represent critical steps [81]. In addition, triterpene bioavailability varies when they are administered as pure isolated compounds or as dietary supplements, for example, contained in olive oil [81]. In the case of using these compounds as isolated therapeutic agents, it is necessary to develop alternative formulations to improve their pharmacokinetic profile and safety and thus increase their therapeutic/toxic ratio. For instance, the administration of ursolic acid (50 mg/kg; p.o.) microcrystals and nanocrystals improve its bioavailability in rats [82]. In addition, nanocrystallization of oleanolic acid (25 mg/kg; p.o.; every alternate day during 7–12 weeks) slightly improves the antioxidant properties of this molecule in rats fed a high fat and fructose diet [33].

Clinical trials on the health effects of triterpenic acids contained in olive oil are scarce. In a meta-analysis of different animal studies (*n* = 23) and a human clinical trial (*n* = 1), which investigated the effects of oleanolic acid formulations on parameters concerning insulin resistance and metabolic syndrome, the authors concluded that oleanolic acid improves blood pressure levels, hypertriglyceridemia, hyperglycemia, oxidative stress, and insulin resistance [17]. Similar results were obtained in a clinical study in patients (30–60 years; *n* = 24) suffering from metabolic syndrome [83]. Treatment with ursolic acid (150 mg/day) for 12 weeks led to a partial (50% of treated patients; *n* = 6) decrease in the body weight, body mass index, waist circumference, and improved insulin sensitivity and fasting glucose levels compared to the placebo group [83]. A daily dose (30 mL) during 3 weeks of virgin olive oils enriched with phenolic compounds and triterpenic acids induced an increase in plasma high density lipoprotein cholesterol and a decrease in plasma endothelin-1 levels in a randomized, crossover, controlled, double-blind, intervention study involving 58 healthy adults [84]. Endothelin-1 levels were also reduced in ex vivo experiments with whole blood cell cultures challenged with phytohaemagglutinin, lipopolysaccharide, and phorbol 12-myristate 13-acetate plus ionomycin [84]. Nevertheless, as Fernández-Aparicio and co-authors suggested, we still have a long way to go, as it is necessary to perform further animal studies that take into consideration all parameters involved in the development of metabolic syndrome to gain a more in-depth understanding of the oleanolic acid effects [17]. In addition, evaluation of the efficacy and safety of long-term intake of olive oil triterpenic acids and whether their effects are transient should be a priority. Altogether, this information would be crucial to create the foundations to perform properly designed clinical trials to evaluate the potential therapeutic use of oleanolic acid in humans.

## 4. Conclusions and Future Perspectives

Pentacyclic triterpenes contained in olive oil, specifically oleanolic acid, are important nutraceuticals with significant pharmacological potency. In this review, we provide information on potential strategies and molecular mechanisms for the treatment of obesity and metabolic syndrome through the therapeutic use of olive oil-derived triterpenic acids, paying particular attention to their signaling and targets related to glucose and insulin homeostasis, lipid metabolism, adiposity, and cardiovascular dysfunction in obesity.

We conclude that the main mechanisms underlying the regulation of body weight, glycemia and insulin sensitivity by oleanolic acid involve regulation of glucose production and uptake, insulin signaling cascades in liver and white adipose tissue, and PPARγ/α activation. In vitro data indicate a potential action of oleanolic acid in insulin secretion by beta-cells, but this evidence needs additional in vivo evidence. The antioxidant effect of oleanolic acid in the liver and adipose tissue is also crucial in the antiobesity effect, by either inhibiting the production of pro-inflammatory cytokines and signaling of the NLRP3 inflammasome or even diverting macrophage infiltration-polarization to the anti-inflammatory M2 phenotype in a mouse model of obesity. In addition, it has been concluded that the cardiovascular effects of triterpenic acids in olive oil significantly contribute to the beneficial effects of these food components in metabolic syndrome. In particular, oleanolic acid is a potent vasodilator that acts mainly through endothelial NO release leading to vasodilation, which is an effective mechanism to ameliorate endothelial dysfunction in pathologies, such as hypertension. Nevertheless, this compound is also able to induce endothelium-independent vasodilatations, and this effect also contributes to its beneficial vascular actions.

Because oleanolic acid can regulate the transcription of genes involved in inflammatory and metabolic pathways in different tissues, it can also be a promising diet-based strategy to promote the resolution of inflammation and the subsequent metabolic disorders associated with obesity. Apart from its antioxidant properties and its ability to regulate mitochondrial function, adipocyte metabolism and secretory function, oleanolic acid is considered a likely candidate for reducing inflammation and oxidative stress associated with obesity.

Subjects with metabolic syndrome are at increased risk of type 2 diabetes mellitus, cardiovascular disease, nonalcoholic fatty liver disease, neurodegenerative disorders, and some types of cancer, due to chronic pathophysiological factors, such as obesity, dyslipidemia, hypertension, and hyperglycemia. Therefore, considering the actions of these olive oil triterpenic acids in the regulation of transcription factors expression, protein kinases, and other molecular targets associated with obesity and related diseases, their incorporation into the diet is a promising strategy to manage the progression of components of obesity-associated diseases, such as insulin resistance and type 2 diabetes mellitus.

Despite these promising preclinical results, further investigations using in vivo models, as well as the development of clinical trials would provide pivotal evidence to better characterize the effectiveness of triterpenic acids in insulin homeostasis, adiposity, inflammation, and oxidative stress in overweight/obese subjects. Moreover, it should be noted that oleanolic acid contained in olive oil has only been evaluated in a single clinical study, while administration as an isolated molecule at clinical stages has not been considered yet. A potential explanation for this shortcoming may rely on its limited bioavailability, the lack of experimental studies that include all the parameters involved in metabolic syndrome, or insufficient information on its safety profile. In general, although preclinical studies have provided promising results and have led to the elucidation of its main signaling pathways, the potential use of oleanolic acid in humans is still far from being firmly established. Therefore, further understanding of oleanolic acid efficacy, molecular targets and safety profile is still much needed for translating research into clinical practice. 

## Figures and Tables

**Figure 1 nutrients-14-01606-f001:**
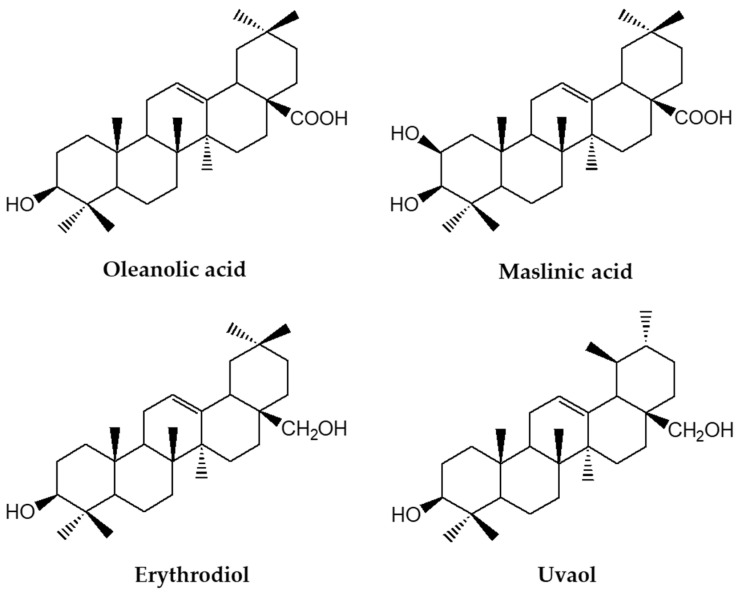
Chemical structures of most important triterpenic acids and alcohols in olive oil.

**Figure 2 nutrients-14-01606-f002:**
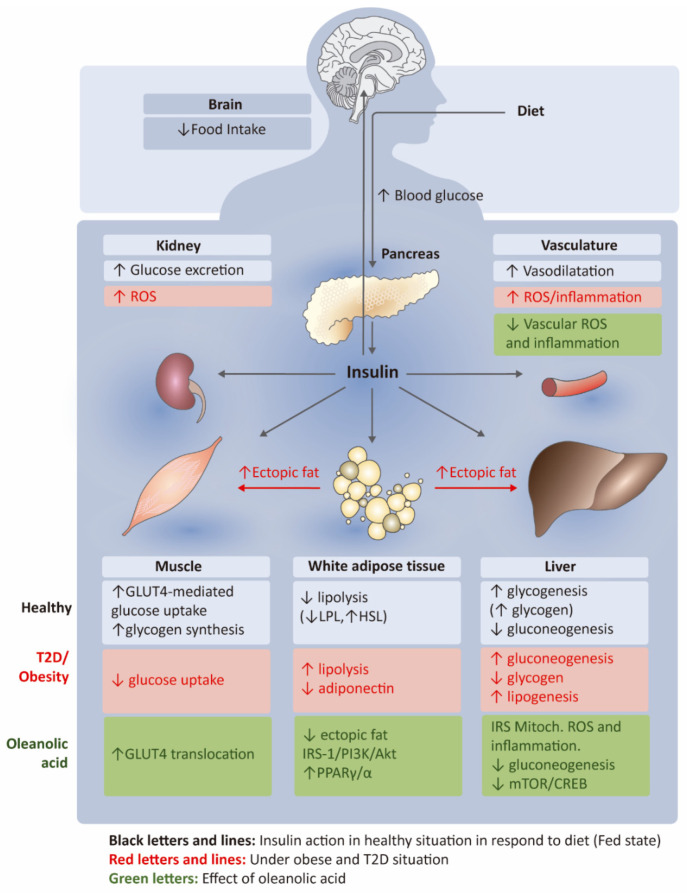
Key actions of insulin under healthy or obesity conditions in central and peripheral tissues in response to diet and the beneficial effects of oleanolic acid. In healthy individuals (black letters), insulin is released in response to diet, leading to glucose uptake in the muscle, hepatic production of glucose and glycogen, and attenuation of lipolysis in white adipose tissue and food intake reduction. These mechanisms are disrupted in obesity and type 2 diabetes (T2D) driving insulin resistance (red letters) with glucose uptake attenuation, increased lipolysis and hepatic production of glucose. In green letters are indicated the main actions and targets by which oleanolic acid is improving insulin sensitivity in obesity, thorough the promotion of glucose transporter translocation in muscle cells, reduction of ectopic fat accumulation, restoration of IRS-, PPAR- and mTOR-dependent pathways, ROS and inflammation in metabolically active tissues in the periphery. ROS, reactive oxygen species; GLUT4, glucose transporter type 4; LPL, lipoprotein lipase; HSL, hormone-sensitive lipase; IRS-1, insulin receptor substrate 1; PI3K, phosphoinositide 3-kinases; Akt, protein kinase B; PPARγ/α, peroxisome proliferator-activated receptor gamma/alpha; IRS-1, insulin receptor substrate 1; Mitoch., Mitochondrial; mTOR, mammalian target of rapamycin; CREB, cyclic adenosine monophosphate (cAMP)-response element binding protein; ↓, decrease; ↑, increase.

**Figure 3 nutrients-14-01606-f003:**
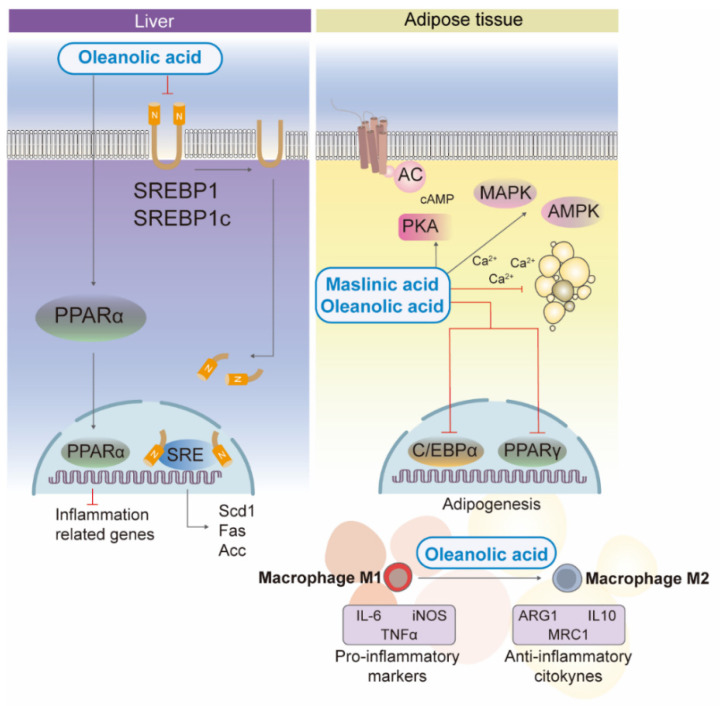
Effects of triterpenic acids on lipid homeostasis and adiposity in liver and white adipose tissue and in brain cells. In the liver, oleanolic acid inhibits the expression of SREBP1-responsive genes and promotes the expression of PPARα leading to anti-inflammatory effects in animal models of obesity. In the adipose tissue, oleanolic and maslinic acids interfere in different steps of adipogenesis and lipolysis by down-regulation of the expression of C/EBPα and PPARγ, and by modulating anti-adipogenic pathways and kinases (MAPK, AMPK, and cAMP-dependent PKA). Oleanolic acid also attenuates inflammation in adipose tissue by diverting macrophage infiltration and polarization to the anti-inflammatory M2 phenotype. SREBP1, sterol regulatory element-binding protein 1; SREBP1c, sterol regulatory element-binding protein-1c; SRE, sterol regulatory element; Scd1, stearoyl coenzyme A desaturase 1; Fas, fatty acid synthase; Acc, acetyl-coenzyme A carboxylase; AC, adenylyl cyclase; cAMP, cyclic adenosine monophosphate; PKA, protein kinase A; MAPK, mitogen-activated protein kinases; AMPK, adenosine monophosphate activated protein kinase; Ca^2+^, calcium ions; C/EBPα, cytosine-cytosine-adenosine-adenosine-thymidine (CCAAT) enhancer-binding protein alpha; M1 and M2, phenotype macrophages; IL-6, interleukin 6; iNOS, inducible nitric oxide synthase; TNFα, tumor necrosis factor alpha; ARG1, arginase 1; IL10, interleukin 10; MRC1, mannose receptor C-type 1.

**Figure 4 nutrients-14-01606-f004:**
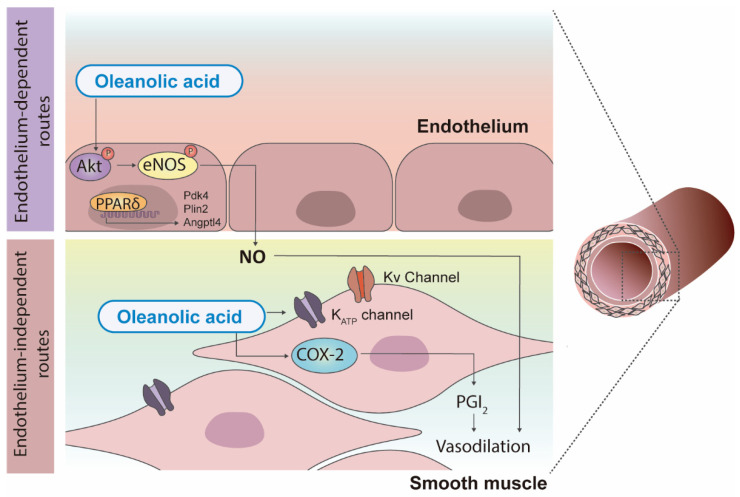
Main mechanisms involved in the beneficial vascular effects of triterpenic acids. Vasodilation evoked by oleanolic acid implies endothelium-dependent and independent pathways. In the endothelium, oleanolic acid induces PI3K-dependent phosphorylation of Akt-Ser (473) followed by phosphorylation of eNOS-Ser (1177) with subsequent release of nitric oxide (NO) and PPARδ activation. In the vascular smooth muscle, oleanolic acid leads to vasodilation throughout activation of ATP-dependent and voltage-activated K^+^ channels and by induction of prostacyclin (PGI_2_) release after COX-2 activation. P, phosphorylation; eNOS, endothelial NO synthase; PPARδ, peroxisome proliferator-activated receptor delta; Pdk4, pyruvate dehydrogenase kinase 4; Plin2, perilipin 2; Angptl4, angiopoietin like 4; Kv, voltage-gated potassium channel; K_ATP_, adenosine 5’-triphosphate (ATP)-sensitive potassium; COX-2, cyclooxygenase-2; Ser, serine.

## Data Availability

Not applicable.

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
