# Peer review of "Molecular Mechanisms Underlying the Effects of Olive Oil Triterpenic Acids in Obesity and Related Diseases"

_nutrients, 2022, doi:10.3390/nu14081606_

Round 1
Reviewer 1 Report
This review summarizes the health benefits of triterpenoids in olive oil. The summary of the molecular mechanisms of triterpenoids is very useful. In addition, information on the bioavailability of each triterpenoid is provided.
It would be more helpful to the reader if specific values of bioavailability (how much triterpenoids were ingested into the body) were provided.
Author Response
We would like to thank the Reviewer for the time taken to prepare the comments. As suggested, we have now incorporated the values of bioavailability (Page 11, section 3, highlighted in red).
In addition, we are now providing a more detailed description of the figures on the captions, since we consider it would be more helpful for the reader. Figure 3 has been corrected (PPARα in liver). All the changes are highlighted in red.
Reviewer 2 Report
This paper by Claro-Cala and collaborators represents a nice review on the molecular mechanisms that underlay the effects of olive oil triterpenic acids in obesity and oher obesity-related diseases. The paper is easy to read and very well-written. Most of the data regard experimental evidences from cellular and animal (rodent) models, while almost no data are available from human to date. The presented figures have a didactic organization which helps a lot in summarizing the presented data. The review is organized with an introductory chapter on the state-of-the-art and in following chapters mainly focusing on a) glucose homeostasis and insuline resistance; b) lipid homeostasis, adiposity and inflammation; and c) cardiovasuclar alterations associated with obesity.
Major points
None
Minor points
None
Author Response
We thank the Reviewer for the time taken to revised the manuscript and for the positive comments. We are now providing a more detailed description of the figures on the captions, since we consider it would be more helpful for the reader. Figure 3 has been corrected (PPARα in liver). All the changes are highlighted in red.